# Development of Fatigue Life Model for Rubber Materials Based on Fracture Mechanics

**DOI:** 10.3390/polym15122746

**Published:** 2023-06-20

**Authors:** Xingwen Qiu, Haishan Yin, Qicheng Xing, Qi Jin

**Affiliations:** 1College of Electromechanical and Engineering, Qingdao University of Science and Technology, Qingdao 266100, China; q936838338@163.com (X.Q.); qichengxing1997@126.com (Q.X.); 2Tongli Tire Co., Ltd., Jining 272100, China

**Keywords:** tire rubber, fatigue damage, numerical simulation

## Abstract

In this paper, the research on the fatigue damage mechanism of tire rubber materials is the core, from designing fatigue experimental methods and building a visual fatigue analysis and testing platform with variable temperature to fatigue experimental research and theoretical modeling. Finally, the fatigue life of tire rubber materials is accurately predicted by using numerical simulation technology, forming a relatively complete set of rubber fatigue evaluation means. The main research is as follows: (1) Mullins effect experiment and tensile speed experiment are carried out to explore the standard of the static tensile test, and the tensile speed of 50 mm/min is determined as the speed standard of plane tensile, and the appearance of 1 mm visible crack is regarded as the standard of fatigue failure. (2) The crack propagation experiments were carried out on rubber specimens, and the crack propagation equations under different conditions were constructed, and the relationship between temperature and tearing energy was found out from the perspective of functional relations and images, and the analytical relationship between fatigue life and temperature and tearing energy was established. Thomas model and thermo-mechanical coupling model were used to predict the life of plane tensile specimens at 50 °C, and the predicted results were 8.315 × 10^5^ and 6.588 × 10^5^, respectively, and the experimental results were 6.42 × 10^5^, with errors of 29.5% and 2.6%, thus verifying the accuracy of thermo-mechanical coupling model.

## 1. Introduction

### 1.1. Theory of Fatigue Research

Tires, seals, shock absorbers and other rubber-based materials will work to produce deformation, long-term exposure to alternating loads will lead to performance degradation, that is, fatigue damage. It is known that fatigue damage accounts for 80% of the failure of rubber products, so it is important to study the fatigue of rubber materials to improve their fatigue performance.

The fatigue damage of rubber materials is divided into two stages: the first stage is the gathering and sprouting of tiny defects inside the rubber to produce tiny cracks, i.e., the crack sprouting stage; the second stage is the expansion of the tiny cracks produced in the previous stage until the rubber material fails by fracture, i.e., the crack expansion stage [1]. Scholars classify the research methods into crack emergence and crack extension methods according to the two stages. The theory of fracture mechanics suggests that there are inherent defects in rubber materials, and when reinforcing materials are added to rubber products, the agglomeration of fillers also causes small defects inside the rubber, and this defect is the source of crack extension. Glanowski et al. [2] used X-ray computed microtomography to observe the fatigue of carbon filled natural rubber and came up with two damage mechanisms: one is the cavitation phenomenon at the poles of the agglomerates, and the other is fracture of the agglomerates, and Huneau et al. [3] studied the fatigue cracking of carbon filled natural rubber and found that the crack initiation mechanisms of carbon black aggregates and oxide aggregates were different, and only the cracks initiated by carbon black aggregates were accompanied by crack extension, because carbon black has stronger cohesion and adhesion to the matrix, and its cohesion is stronger than adhesion, so the crack budding is actually also the crack Therefore, it is of scientific significance to apply fracture mechanics to study the fatigue process of rubber materials.

According to fracture mechanics, when rubber cracks grow, a new surface will be generated, and the generation of a new surface will inevitably consume energy, that is, surface energy. When the mechanical energy storage consumed by the crack per unit area of expansion is greater than this re-sistance, the crack will expand [4]. The tearing energy is defined as the mechanical storage energy *dU* required to produce a crack per unit area *dA* [5], in which:(1)$$T=-\frac{dU}{dA}$$
where the negative sign indicates that the strain energy required to produce cracks in rubber materials decreases as the crack area increases.

The study of rubber fatigue life dates back to 1940, when Cadwell [6] studied the dynamic fatigue life of natural rubber, followed by Thomas [7], Lake [8], Lindley [9], Rivlin [10], and William V. Mars [11], which was developed over the years to form a whole set of theories, namely the crack extension theory based on fracture mechanics. They combined a large number of experimental data of crack extension under transverse loading, and divided the crack tip energy release rate-crack growth rate into four regions according to the maximum energy release rate in the cycle, as shown in Figure 1 [12], $T_{0}$ indicates the threshold tear energy, and the peak tear energy of stage I, $T_{max}$ ≤ $T_{0}$, at this stage the crack will not expand due to the external load; $T_{t}$ is the transition tear energy, which indicates the transition from stage II to stage $T_{t}$ is the transition tear energy, which indicates the tear energy corresponding to the transition from stage II to stage III; $T_{c}$ indicates the critical tear energy, when $T_{max}$ ≥ $T_{c}$, the crack reaches stage IV, i.e., the destabilization expansion process. The approximate expressions for the crack growth rates of the four stages are also given, as follows
(2)$$\left\{\begin{array}{ll} \frac{da}{dN}=r_{z} & T_{max}<T_{0}\\ \frac{da}{dN}=A_{0}\left(T_{max}-T_{0}\right)+r_{z} & T_{0}\leq T_{max}<T_{t}\\ \frac{da}{dN}=BT_{max}^{F} & T_{t}\leq T_{max}<T_{c}\\ \frac{da}{dN}=\infty & T_{max}=T_{c} \end{array}\right.$$
where $r_{z}$ is the crack extension rate when the tear energy is less than the threshold tear energy, and $A_{0}$, *B*, and *F* are the constants associated with the rubber material.

Among the above four stages, stage II and stage III belong to the stable expansion stage, and only the crack expansion rate of stage III satisfies the power exponential relationship with the peak tear energy, and this is also the stage in which the rubber products are often in when fatigue problems arise, so stage III can be used instead of the actual crack expansion process.

The visual representation can be obtained by taking the logarithm of both sides of the equation of stage III at the same time
(3)$$\log\frac{da}{dN}=\log B+F\log T_{max}$$

Therefore, this equation becomes a straight line in logarithmic coordinates, and the influence of various factors on the crack extension of the rubber material can be shown more intuitively by the slope. Moreover, by establishing a fatigue life model to evaluate the crack expansion law, the data distortion caused by one experiment can be avoided.

After the crack extension rate is measured by the experimental platform built in this paper, it is also necessary to know the maximum tear energy density at the current strain level, and the equation of tear energy density is shown in Equation (4):(4)$$T_{max}=2k(\lambda )l_{0}E_{0}$$
where is the strain energy density of the rubber specimen without pre-crack, is the length of the pre-crack, and is a function of the strain related to the rubber specimen, which is calculated as
(5)$$k\left(\lambda \right)=\frac{\pi }{\sqrt{\lambda }}$$
where is the elongation of the test piece.

Substituting Equation (5) into (4) yields
(6)$$T_{max}=\frac{2\pi lE_{0}}{\sqrt{\lambda }}$$

Combining Equations (3) and (6) yields
(7)$$N=\int_{l_{0}}^{l_{c}}B^{-1}T^{-F}da=\frac{1}{B\left(F-1\right){\left(\frac{2\pi }{\sqrt{\lambda }}E_{0}\right)}^{F}}(\frac{1}{l_{0}^{F-1}}-\frac{1}{l_{c}^{F-1}})$$
where, is the length of the preset crack, is the length of the crack at fatigue failure. The Formula (7) is the formula for calculating the fatigue life of rubber materials.

### 1.2. The Effect of Temperature on Rubber Fatigue

Most rubber products are used in air and therefore most rubber fatigue studies are in air medium. Le-Gorju et al. [13] found that rubber exposed to air reacts with oxygen, especially at high temperatures where oxidation is more intense. Temperature is a crucial factor affecting the fatigue life of rubber, which is a temperature-sensitive material, and different types of rubber have different sensitivity to temperature; for example, the life of natural rubber at 110 °C is 1/4 of that at 0 °C, while the life of butadiene rubber decreases by 10,000 times [14]. Ruellan et al. [15] conducted uniaxial tensile fatigue experiments on filled natural rubber at different temperatures and found that strain crystallization was clearly observed on the fracture surface at room temperature, while at 90 °C, a significant decrease in crystallization could be observed on the fracture surface, and the crystallization disappeared completely at 110 °C. Therefore, high temperature affects the crystallization of natural rubber; Ngolemasango et al. [16] also found such a situation in their experiments. Rey et al. [17] studied the change in the properties of silicone rubber at different temperatures, and when the microstructure stable, the hardness of unfilled silicone rubber increased continuously with increasing temperature; while for filled silicone rubber, its hysteresis, stress relaxation and stress softening decreased continuously with increasing temperature. Haroonabadi et al. [18] thermally aged allene nitrile butadiene rubber (NBR) for 7 days and found that its crosslink density increased while its tensile and tear strength decreased. Chou et al. [19] thermally aged EPDM rubber for 6 months and then conducted fatigue experiments, which showed that the rubber life was reduced regardless of whether it was filled with carbon black. This is due to the fact that high temperature accelerates the thermal oxygen reaction, which continuously degrades the cross-linked network, so the increase in aging temperature and time irreversibly reduces the fatigue life of the rubber [20]. Luo et al. [21] conducted fatigue experiments on hourglass-type rubber specimens, in which real-time monitoring of the surface temperature revealed that the surface temperature remained in a stable interval for a long time, which accounted for most of the fatigue life. Then, the surface temperature increases sharply when the specimen is close to destruction, so the change in temperature can be used to determine when the specimen is reaching its fatigue limit. Then, the relationship between the steady-state temperature rise and the maximum principal strain is determined so as to determine the fatigue life of rubber.

The most important feature of natural rubber that distinguishes it from synthetic rubber is the crystallization phenomenon, which includes strain-induced crystallization and low-temperature-induced crystallization, and a large number of studies have shown that the fatigue life of natural rubber is substantially enhanced under non-relaxation loading (i.e., the ratio of minimum loading to maximum loading > 0) [22]. The crystallization phenomenon is considered to be the reason for the high fatigue resistance of rubber materials. Rubber materials exhibit different properties at different temperatures, for example, below the glass transition temperature, rubber is in a glassy state, and above that temperature, it is in a highly elastic state, indicating that rubber materials are highly temperature-sensitive materials. Most rubber products work in environments ranging from ambient temperature to 100 °C. Federico et al. [23] showed that microdefects in rubber materials at ambient temperature mainly originate from cavities generated by the separation of agglomerates from the rubber matrix, while at high temperature, they mainly originate from the fracture of agglomerates. Schieppati et al. [24] studied the fatigue properties of NBR, and found that the higher the temperature, the faster the crack growth rate, while the crack growth rate changed slightly at 25 °C~40 °C. Liu Xiangnan et al. [25] conducted uniaxial tensile fatigue tests on dumbbell-shaped natural rubber specimens, and found that the fatigue life was dispersive under the same conditions, so three probability distribution models, namely normal distribution, lognormal distribution and Weibull distribution were used to quantify the life distribution. Demiral et al. [26] made a finite element simulation of a bonded joint and used the user-defined UMAT subroutine of ABAQUS/Standard to link the static damage and fatigue damage models of the bonded zone to express the response of the bonded layer. Fang Yunzhou et al. [27] adopted the Arrhenius model, introduced the high temperature aging factor into the fatigue model with the engineering strain peak as the damage parameter, and accurately predicted the fatigue life of rubber bushing using finite element analysis. Weng et al. [28] subjected natural rubber to a high temperature (85 °C) and cyclic loading at the same time. SEM images confirmed the appearance of nano-scale cracks and cavities under the combined action of high temperature and cyclic loading. Unlike the fatigue loading conditions at room temperature, the cracks were caused by the nucleation effect of dissolved steam and gas in the low-molecular-weight domain of NR. The appearance of a low-molecular-weight domain is caused by thermal degradation products. Zhang et al. [29], based on the uniaxial tensile fatigue data of dumbbell-shaped rubber specimens, used the least square method to fit the functional relationship between the strain energy and temperature, established the high-temperature life-prediction formula and achieved good results. Luo et al. [30] studied the static tearing behavior of carbon-black-filled rubber at different temperatures and measured the critical tearing energy. The results showed that the critical tearing energy decreased exponentially with the increase of temperature.

### 1.3. Research Significance

The fatigue performance of rubber is affected by many factors, among which the temperature is the most significant. Many factors that affect the fatigue performance of rubber actually affect its fatigue by changing the temperature of rubber. For example, frequency itself has no effect on fatigue performance, and high frequency increases the temperature of rubber, thus affecting its fatigue life. Seichter et al. [31] reached the same conclusion when studying the effect of frequency on rubber fatigue. Therefore, the study of rubber fatigue should be based on temperature and establish the analytical relationship between temperature and fatigue life. At present, there are few studies on fatigue prediction of rubber based on crack propagation method, and no unified conclusion has been formed.

The main purpose of this paper is to build a fatigue experiment platform and design a fatigue experiment method based on the crack propagation method, so as to form a set of evaluation methods for studying rubber fatigue at variable temperature, and then establish a rubber thermal-mechanical coupling fatigue prediction model according to a large number of experimental results, which provides valuable reference for rubber fatigue research. Figure 2 is the technical route of research.

## 2. Construction of Fatigue Test Rig and Parameter Acquisition

### 2.1. Mullins Effect Study

When the rubber undergoes such a cyclic loading of stretching-unloading-restretching, the stresses of unloading and restretching are smaller than those of stretching, which is the Mullins effect of rubber, also known as stress softening phenomenon, and is a special mechanical property in rubber, especially in filled rubber, which has an important influence on the acquisition of stress-strain experimental data. This experiment was conducted to study the Mullins effect at different stretching speeds, starting from 2 mm/min and increasing to 500 mm/min, which were divided into low-speed stretching and high-speed stretching in order to distinguish them. In order to minimize the effect of speed during unloading in cyclic loading experiments, the unloading speed was unified to 2 mm/min when low-speed experiments were conducted, and the unloading speed was set to 50 mm/min under high-speed experiments. low-speed experiments were set to stretch at 2 mm/min, 5 mm/min, 10 mm/min, 20 mm/min; high-speed experiments were set to stretch at 50 mm/min, 100 mm/min, 500 mm/min.

#### 2.1.1. Low-Speed Stretching (≤50 mm/min), the Effect of Stretching Speed on the Mullins Effect

In this experiment, experimental tests were conducted to stretch 12.5 mm, 25 mm, 37.5 mm, 50 mm, and 62.5 mm, i.e., strain levels of 50%, 100%, 150%, 200%, and 250% to study the effect of different stretching speeds on the Mullins effect, and the experimental data for 50% and 150% strains were selected in this case, as shown in Figure 3 and Figure 4. It can be seen that the faster the stretching speed is, the higher the required stress is when stretching to the maximum strain, whether it is 50% or 150% strain; and it can be seen that the stretching curve for the larger stretching speed is above the curve for the lower speed.

In this experiment, experimental tests were conducted to stretch 12.5 mm, 25 mm, 37.5 mm, 50 mm, and 62.5 mm, i.e., strain levels of 50%, 100%, 150%, 200%, and 250% to study the effect of different stretching speeds on the Mullins effect. The experimental data for 50% and 150% strain were selected in this case, as shown in Figure 5 and Figure 6. It can be seen that the faster the stretching speed is, the higher the required stress is when stretching to the maximum strain, whether it is 50% or 150% strain. It can also be seen that the stretching curve for the higher stretching speed is above the curve for the lower speed.

This can be explained by the high elasticity deformation theory of rubber [32]. The molecular structure of rubber creates a special property of high elasticity, where many elongated molecular chains are either adsorbed on the filler particles or in a free curl, with a large amount of flexibility and mobility. When the rubber is subjected to tensile load, the molecular chains are gradually stretched from the initial free-curl state so that the molecular chains are oriented and therefore the conformational entropy of the molecular chains decreases. From the thermodynamic point of view [33], the tensile load transmits mechanical energy to the rubber molecules, and this energy is converted into the thermal motion of the rubber molecules, which induces the rubber molecular chains to return from the stretched state to the free-curl state, increasing the conformational entropy. This is why the rubber material can return to its original length after intense deformation. The greater the stretching speed, the greater the strain rate, the higher the frequency, the more intensified the thermal movement of molecules, and the more high-speed stretching in a very short period of time. This means that the high-speed stretching process is somewhat adiabatic, that is, the heat generated by the thermal movement of molecules cannot be dissipated and can only increase, so it accelerates the increase in conformational entropy.

#### 2.1.2. High-Speed Stretching (≥50 mm/min), the Effect of Stretching Speed on the Mullins Effect

As shown in Figure 5, the cyclic stretching curve at a high speed is similar to that at a low speed.

And it can also be seen that the curve has a tendency to rise suddenly at 500 mm/min stretching speed, which may be a sudden increase in stress caused by the strain crystallization of natural rubber.

As shown in Figure 6, the cyclic stretching curves are plotted for stretching speeds of 50 mm/min, 100 mm/min and 500 mm/min, respectively, and it can be seen that the rising curve (c) has a sudden upward trend at the beginning of loading, which also proves the possibility of strain crystallization of natural rubber as described above. At the same time, it can be seen that the unloading curve of each cyclic stretching is steeply decreasing, which indicates that the cohesive structure inside the rubber is destroyed in a large amount, and the stress required for unloading decreases after each destruction, and the loading curve of the latter stretching is suddenly increasing near the top of the loading curve of the previous stretching, which also reflects the continuous destruction of the cohesive structure inside the molecule. From the thermodynamic point of view, the loading process stretches the molecular chain to the straightened state at each cycle of stretching, and the conformational entropy decreases accordingly, and the rubber deformation decreases when unloading, but the molecular chain does not return to the curled state completely, and the conformational entropy does not decrease again from the equilibrium state when loading again, but decreases from below the equilibrium state, so the required stress decreases when stretching again.

An explanation for the Mullins effect was given by Diani and Marckmann et al. [34,35], who explained the Mullins effect in terms of chain breakage, where they suggested that the rubber matrix reacts with the filler by cross-linking during vulcanization to form a cross-linked structure, and the applied load leads to the stretching and breaking of the molecular chains.

As shown in Figure 7, (a) is the state of the molecular chain when the rubber is not subjected to tensile load, and the molecular chains A, B and C are adsorbed on the filler particles through chemical adsorption and physical adsorption, and are in the bent state; (b) is the state of the rubber specimen when it is stretched. When the stretching experiment is carried out, the internal molecular chains of the rubber start to be stretched, firstly the shorter molecular chain C is stretched from the bent state to the straightened state, at this time the straightened molecular chain bears most of the stress and continues to stretch, the longer molecular chain A is also gradually stretched from the bent state to the straightened state, while the shorter molecular chain C is pulled off due to overload. This is the reason why the stress-strain curve of the re-stretching in the above cyclic stretching experiment is always lower than the stress-strain curve of the first stretching.

The Mullins effect has an impact on the acquisition of rubber material parameters, so it is necessary to eliminate the Mullins effect of rubber materials when testing material parameters, as shown in Figure 8. The parameters of the rubber materials were first stretched in five cycles, and the data of the sixth stretching were taken.

From the above experiments on the effect of stretching speed on the Mullins effect, it is concluded that the damage to the internal structure of the rubber material is different when stretched to different strain levels, and when subjected to smaller strains, the filler network inside the rubber is damaged first, and when the strain level continues to increase, the reinforcement network generated by the rubber matrix and the filler starts to be damaged, and when stretched to a larger strain level, the interpolymer The network structure between the polymers also starts to be destroyed when the strain level is increased [36]. At the same time, this experiment also leads to another conclusion that the tensile experiments of rubber materials cannot be simply classified as static or dynamic, because the boundary of distinction is blurred, and this experiment was conducted by varying the tensile speed, and it was concluded that as the tensile speed increases, the stress required to stretch the rubber material to the same strain level also increases, which is due to the fact that the main unit of motion of rubber in the high-elastic state is “When increasing the stretching speed, the strain rate of rubber molecules increases, and the smaller “chain segments” are first subjected to tension and are the main units of motion, so the stress required for stretching is greater.

### 2.2. Fatigue Test Bench Construction

From the previous section, it is known that the crack expansion stage can be divided into four stages, and the third stage is the focus of calculation. To accurately calculate the fatigue life, accurate material parameters should be obtained, and as a polymer material capable of withstanding intense deformation and with nonlinear characteristics, an accurate description of its constitutive model is the key to calculate its fatigue life. From previous studies, it is necessary to fit the experimental data of uniaxial tensile, equiaxial tensile and plane tensile to describe the hyperelastic constitutive model of rubber, and the loading schematic is shown in Figure 9. These three experiments characterize the mechanical behavior of rubber materials in uniaxial tension, uniaxial compression, and pure shear states, respectively, and the alternating loads on rubber products in actual use can be represented by the coupling of these three force states [37].

From the mechanical point of view, The fatigue test in this paper is shear fatigue, and there is no equal biaxial tensile stress field. Only the experimental data of uniaxial tension and planar tension are needed to fit the hyperelastic constitutive model of rubber more accurately.

From Figure 9, it can be seen that the stress field of the rubber material is different for different specimen shapes and tensile states, and the rubber specimen required for uniaxial tensile is a flat dumbbell specimen, and the flat tensile specimen is self-designed, as shown in Figure 10. Mechanically, when the length-width ratio of the specimen is greater than or equal to 10, it can be approximately considered that the stress state is in a pure shear state, so the size of the specimen is designed to be 140 mm long, the height of the working area is 10 mm, and the thickness is 2 mm. For fatigue test, it is necessary to preset a crack of 25 mm, as shown in Figure 11.

In the uniaxial tensile experiments, the Mullins effect has a greater impact on the acquisition of experimental data, which can be essentially eliminated after 5–6 times of stretching, so the stress–strain data from the sixth stretch is selected after five cycles of stretching first. However, it should be noted that the creep phenomenon also exists in rubber materials, so after five times of cyclic stretching, the equipment should be allowed pause for 1 min so that the creep phenomenon can be alleviated, then the extensometer should be reclamped to maintain the standard distance and straightened state to ensure the length of 25 mm, and then the sixth stretching can be carried out.

The equipment used for uniaxial stretching of rubber is the universal experiment machine of high speed rail and its own fixture. The plane stretching experiment requires self-designed fixture for the experiment because there is no national standard, and the fixture is shown in Figure 11. To ensure the accuracy of the experimental data, each test was conducted three times and the average value was taken. The process of obtaining tensile stress-strain data is shown in Figure 12, with a tensile speed of 50 mm/min first 5 cycles of tensile to eliminate the Mullins effect, and the sixth tensile to obtain stress-strain data after resting for 1 min.

In this paper, based on the relationship between the crack expansion rate and tearing energy established by Thomas et al., the fatigue life calculation model of rubber materials was established in order to obtain the material parameters required for calculating the fatigue life of rubber using the crack expansion method. We designed the fixture, specimen and vulcanization mold used in the shear fatigue experiment of rubber materials and built the planar tensile and fatigue experimental platform according to the characteristics of fatigue experiments, as shown in Figure 13a. Since the conventional fixture is prone to screw loosening after a long period of fatigue experiments, which leads to the problem of the specimen clamping sliding and eventually distorting the stress–strain data during the test, the new fixture and specimen used in this paper can avoid the above problems. These two reasons are likely to lead to errors in the experimental results and affect the accuracy of the subsequent life calculation process. Therefore, after considering various factors, we designed our own fixture and specimen. The designed specimen is vulcanized into a cylindrical shape at the upper and lower ends, and the upper and lower two cylinders are embedded in the cylindrical groove of the designed fixture. The computer can display the tensile stress in real time, and adjust the stress to 0, that is, to reach the initial unstressed state of the specimen, after which the experiment can begin. This fixture can avoid the clamp loosening caused by the screw becoming loose accurately capture any quick changes in any part of the specimen, and also avoid errors caused by the inaccurate height of the single measurement work area. When carrying out fatigue test, a 25 mm crack is preset on one side of the specimen, as shown in Figure 13b, According to fracture mechanics, when the crack length is greater than or equal to twice the height, the influence of edge effect on the shear fatigue test can be ignored. Additionally, the accuracy when measuring the crack expansion length affects the accuracy when constructing the fatigue life calculation model, so this test bench is designed with a vision system for measuring the crack expansion length, as shown in Figure 13b.

### 2.3. Selection of Constitutive Model

This experiment is conducted as a shear fatigue experiment, and its stress field is a combination of uniaxial and planar tension, so fitting the intrinsic parameters requires the use of stress-strain data from uniaxial and planar tension, however, some literature uses only data from uniaxial tension experiments for fitting, and the accuracy of the fitting results needs to be considered. Assuming that the rubber deformation is within the Gaussian chain and conforms to the Gaussian distribution function, the strain energy function equation [38],
(8)$$W=\frac{1}{2}G(\lambda_{1}^{2}+\lambda_{2}^{2}+\lambda_{3}^{2}-3)$$
where is the strain energy of the rubber, is the shear modulus, and is the elongation in the x, y, and z directions, respectively.

In uniaxial stretching, the elongation in the three directions are *λ*1 = *λ*, *λ*2 = *λ*3 = *λ* − 1/2, so the (8) equation becomes
(9)$$W=\frac{1}{2}G(\lambda^{2}+\frac{2}{\lambda }-3)$$

Derivation of Equation (9) yields
(10)$$\sigma =G(\lambda -\frac{2}{\lambda^{2}})$$

Equation (10), which is the equation of state for uniaxial stretching of rubber materials;

Similarly, the equation of state for plane stretching is obtained for the state of *λ*1 = *λ*, *λ*2 = 1 and *λ*3 = *λ* − 1.
(11)$$\sigma =G(\lambda -\frac{1}{\lambda^{3}})$$

Isobiaxial stretching with *λ*1 = *λ*2 = *λ* and *λ*3 = *λ* − 2 gives the equation of state for isobiaxial stretching
(12)$$\sigma =2G(\lambda -\frac{1}{\lambda^{5}})$$

From the state equations of the above three stress fields, it can be seen that the specimens in the same elongation ratio have the highest stress when subjected to equal biaxial stretching, followed by planar stretching, and the lowest is seen in uniaxial stretching. The gap between them can be seen more clearly from the experimental graph.

As shown in Figure 14 for the experimental data and simulation data of uniaxial stretching and plane stretching, it can be seen that there is a large gap between the stress-strain curves of uniaxial stretching and plane stretching, and the hyperelastic model needs to be used in calculating the strain energy density of rubber, so for the stress field of the combination of uniaxial stretching and plane stretching, the data of uniaxial stretching cannot be used only, which will cause large deviation, so it is necessary to The uniaxial tensile and planar tensile data need to be obtained simultaneously. In this paper, we use Ogden model to fit the experimental data with a high degree of overlap with the experimental data, which can meet the calculation requirements. The results are shown in Table 1.

The non-working area of the specimen in this experiment is cylindrical, which is likely to cause convergence problems for the finite element simulation, and deleting this part during the simulation will not affect the simulation results, so the non-working area is removed from the simulation. As shown in Figure 15, the strain energy density is required in the calculation model of ABA fatigue life, and the strain energy density is calculated by finite element simulation in this paper. The size of the rubber fatigue specimen in this experiment is 140 mm × 30 mm × 2 mm, and the size of the working area is 140 mm × 10 mm × 2 mm. As the finite element model is established in QUS, the material parameters are set and submitted for calculation. The maximum strain energy density of the working area is obtained. As shown in Figure 15, the grid division diagram of the finite element model is shown, and Figure 16 shows the loading mode.

There is an edge effect in the calculation of the finite element model, as shown in Figure 17b, which needs to be removed manually, so the maximum strain energy density should be selected after removing the value of the fixture edge. The strain energy densities at different strain levels are shown in Table 2. The maximum tear energy Tmax can be obtained by substituting the maximum strain energy density under different strains into Equation (4).

## 3. The Relationship between Fatigue and the Variables

Rubber is a temperature-sensitive polymer material, and different working conditions will have different effects on its fatigue performance, especially the temperature. High temperature will make the rubber material soft, so the stress-strain curve at high temperature will also be different. This paper intends to investigate the effects of different factors on the fatigue performance of rubber by conducting fatigue experiments at different temperatures, different frequencies, different orientations and different loading methods.

### 3.1. Stress Ratio

The stress ratio R is the ratio of the minimum stress to the maximum stress that the rubber specimen is subjected to during cyclic loading. For different rubber materials, the effect of stress ratio on fatigue life is different. Scholars at home and abroad have conducted a large number of experiments to study the effect of stress ratio, and although some useful conclusions have been drawn, the overall law is not universal, so to understand the effect of stress ratio of the formulation used in this experiment, it needs to be studied by experiment.

Poisson et al. [39] conducted uniaxial tensile fatigue experiments using neoprene and showed that when the stress ratio R ≥ 0.2, the fatigue life of the specimen increased with the increase of the stress ratio, which means that the crack expansion rate was decreasing. In the study of magneto-rheological elastomers, Yong [40] concluded that the life of magneto-rheological elastomers remained almost unchanged as the stress ratio increased, which he attributed to the fact that the fatigue of magneto-rheological elastomers belongs to high circumferential fatigue and the stresses applied during the experiments were much less than their fatigue limits.

While this experiment was conducted with different stress ratios, as shown in Table 3, it was found that the fatigue life of the rubber material surged when the stress ratio was applied for the experiment, and the fatigue life increased by an order of magnitude with the stress ratio, and the fatigue life of the rubber increased with the increase of the stress ratio, which indicates the possibility of crystallization of natural rubber in the tensile state, as observed by Beatty et al. [41] in their study Non-crystalline rubber in the stress ratio R > 0 was not observed to enhance the life; and the rubber material has creep phenomenon, the rubber material creep during long time clamping, as the experiment proceeds, the stress ratio R due to the creep of rubber will become smaller and smaller, so the change in rubber strain caused by the stress ratio R will be smaller and smaller, then the transfer to the crack interface, caused by the damage to the crack tip should be small. This may be the reason why the increase of stress ratio will cause the fatigue life to increase.

### 3.2. Loading Method

As shown in Figure 18, the effect of different loading methods on fatigue life is negligible, and the fitted function curves are
(13)$$\log\left(\frac{dc}{dN}\right)=3.117\log\left(T_{max}\right)-10.571$$
(14)$$\log\left(\frac{dc}{dN}\right)=3.114\log\left(T_{max}\right)-10.561$$
(15)$$\log\left(\frac{dc}{dN}\right)=3.137\log\left(T_{max}\right)-10.614$$

The above Equations (13)–(15) are the fitted curves under triangle wave, square wave and sine wave loading respectively, and it can be seen that the slope and intercept difference of the three equations are very small, because the different loading methods affect the change of strain following stress, and the same material has the same kinematic unit, so the relaxation time of strain relative to stress change is unchanged at the same frequency; meanwhile At the same time, the temperature of the specimen will not rise when loading at low frequency, so it will not produce the change of temperature and thus affect the change of internal energy. These two reasons may be the reason why the loading method does not affect the fatigue life of rubber materials.

### 3.3. Frequency

The effect of frequency on the rubber material is reflected in the fact that high frequency increases the temperature of the rubber material, which affects the crack expansion rate. As shown in Figure 19, fatigue experiments were conducted at 1 Hz, 5 Hz, 8 Hz, 12 Hz, and 15 Hz, respectively, and it can be seen that the change in crack expansion rate at 1 Hz, 5 Hz, and 8 Hz is small, while the crack expansion rate increases significantly at 12 Hz and 15 Hz frequencies. The fitting equations for the five frequencies are shown in the following Equations (16)–(20).
(16)$$\log\left(\frac{dc}{dN}\right)=3.259\log\left(T_{max}\right)-10.803$$
(17)$$\log\left(\frac{dc}{dN}\right)=3.283\log\left(T_{max}\right)-10.845$$
(18)$$\log\left(\frac{dc}{dN}\right)=3.282\log\left(T_{max}\right)-10.831$$
(19)$$\log\left(\frac{dc}{dN}\right)=3.389\log\left(T_{max}\right)-10.948$$
(20)$$\log\left(\frac{dc}{dN}\right)=3.507\log\left(T_{max}\right)-11.128$$

From the first three equations, the slope and intercept difference is very small, while the slope of the last two equations increases significantly, indicating the enhanced crack expansion rate. It indicates that the first three frequencies did not lead to a significant increase in the temperature of the rubber specimen, so the change in frequency did not have an effect on the fatigue life, while the high frequencies led to an increase in the temperature of the rubber, so the internal energy and conformational entropy of the rubber increased, making the generation of new surfaces easier and therefore accelerating the crack extension rate.

### 3.4. Effect of Orientation on Fatigue Life

As shown in Table 4, the fatigue life of the specimens under different orientations has a large difference, and the orientation refers to the direction of the rubber material taken during vulcanization of the rubber specimen. The polymer chain of the rubber material is a long chain linear structure, so the direction of the molecular chain will be affected by the process. Generally, the direction of the molecular chain is the same as the calendering direction, so the way of taking rubber during vulcanization is also very important. If the orientation of the vulcanized specimen is the same as the calendering direction, the modulus and strength of the specimen can be improved; when the direction of the preset crack of the fatigue specimen is the same as the calendering direction, which is equivalent to the tearing direction and the molecular chain direction is parallel to the direction of calendering, and the direction of the preset crack is perpendicular to the direction of calendering, then the molecular chain must be torn off if the crack wants to expand, so the crack expansion rate is lower when the orientation is perpendicular, i.e., the fatigue life is higher.

### 3.5. Effect of Mullins Effect on Fatigue Life

From the crack expansion rates in Table 5, it can be seen that the Mullins effect has some influence on the fatigue life. Of course, the difference between the specimens with and without the Mullins effect eliminated from the crack expansion rate is not large, which can also be considered as being caused by the experimental error. The elimination of the Mullins effect will cause a certain degree of damage since the cyclic tension process will cause the destruction of the filler network, the debonding between the rubber matrix and the filler, etc., which can create internal micro-defects of different degrees in the rubber. However, it cannot be simply assumed that the Mullins effect causes a reduction in fatigue life because the 5–6 cycles of stretching required to eliminate the Mullins effect are negligible compared to the fatigue life of tens of thousands of cycles. But whether it is reasonable to stretch to a higher strain level to eliminate the Mullins effect or whether it has an effect on the fatigue life requires further systematic experiments to verify.

### 3.6. Construction of a Thermodynamic Coupling Model for Fatigue Life of Rubber

The importance of temperature for rubber is self-evident, and scholars started to study the effect of temperature on the fatigue life of rubber materials a long time ago. Mars found that the life of butadiene rubber decreased by 10^4^ and that the life of natural rubber decreased by four times when the temperature increased from 0 °C to 100 °C. Viscoelasticity is one of the characteristics of rubber, and its elastic energy storage is transformed into heat energy when subjected to alternating load for a long time, which increases the temperature of rubber, especially for tire products, and the internal temperature of tires can reach nearly 100 °C when rotating at high speed, so it is of great significance to study the effect of temperature on rubber.

In this experiment, the crack expansion behavior and static tearing behavior of rubber materials are studied by changing the ambient temperature, the critical tearing energy is determined by static tearing experiments at different temperatures, and the critical tearing energy is the criterion on which to judge whether the rubber crack expansion is destabilized. Firstly, the specimens with preset cracks are stretched at a speed of 50 mm/min until they tear, and the state of instantaneous fracture is the strain corresponding to the critical tear energy.

In order to prevent damage to rubber caused by excessive thermal aging, the specimens were preheated at a predetermined temperature for 12 min before each trial, the surface temperature of rubber was measured to reach the predetermined temperature, and then the experiment was started.

As shown in Figure 20, the fatigue life fitting curves at different temperatures are plotted, and the fitting functions are, respectively
(21)$$\log\left(\frac{dc}{dN}\right)=3.019\log\left(T_{max}\right)-10.779$$
(22)$$log(\frac{dc}{dN})=3.157log(T_{max})-10.658$$
(23)$$log(\frac{dc}{dN})=3.106log(T_{max})-10.380$$
(24)$$log(\frac{dc}{dN})=4.020log(T_{max})-11.906$$
(25)$$log(\frac{dc}{dN})=4.263log(T_{max})-12.583$$

Equations (21)–(25) are the fitting functions at 0 °C, 20 °C, 30 °C, 50 °C and 70 °C, respectively. It can be seen that the slope of the curve gradually increases with the increase of temperature, but the slope at 20 °C is larger than that at 30 °C, which means that the crack expansion rate is higher at 20 °C. The possible reason is that the increase in temperature makes the rubber soft, and the stress is smaller when reaching the same strain, so the stress concentration of the crack tip is low, and the failure stress is not reached. This means that crack branching is more likely to occur and that there is more energy dissipation in the generation of new surfaces, therefore hindering the expansion of the crack. The phenomenon of crack branching was also observed in the experiment. Meanwhile, the fitted curve at 0 °C is lower than other that at temperatures, indicating that the rubber material is more prone to crack defects at higher temperatures, thus intensifying crack expansion. While continuing to increase the temperature, the crack expansion rate in the temperature range of 50–70 °C continues to increase, indicating that in this temperature range, the fatigue life of the rubber material decreases sharply due to the high temperature, and more micro-defects are generated inside the rubber when bearing the load. The long time spent bearing the alternating load makes the micro-defects gather and expand, while the temperature increase as the activation energy of the crack tip increases, causing the cracks expand rapidly. From above, it can be seen that high temperatures limit the strain-induced crystallization of rubber, so the life strengthening disappears. The above two aspects may be the reason for the poor fatigue resistance of rubber materials under high temperature.

Therefore, it can be concluded that the fatigue performance of rubber materials with the change of temperature is not a linear increase or decrease, but rather there are three stages: from 0 °C to about 20 °C, the crack expansion rate increases with the increase in temperature; while from 20 °C to 30 °C, temperature instead appears to increase the fatigue life; and when continuing to increase the temperature to between 50 °C and 70 °C, the fatigue life decreases sharply.

It can be seen that temperature has a great influence on the fatigue life of rubber materials, and the working environment of many rubber products is not room temperature, so it is necessary to take the influence of temperature into account when establishing the calculation model. From the above study, it is found that the tearing energy is related to temperature, so the tearing energy is considered as a function of temperature, and from the calculation formula of tearing energy:(26)$$T_{max}=2k(\lambda )l_{0}E_{0}$$

$k\left(\lambda \right)$ is a function related to the strain of the rubber specimen, which is calculated as
(27)$$k\left(\lambda \right)=\frac{\pi }{\sqrt{\lambda }}$$

Substituting (27) into (26) yields
(28)$$T_{max}=\frac{2\pi lE_{0}}{\sqrt{\lambda }}$$

The tearing energy obtained at different temperatures is plotted in Figure 21, and fitting the experimental data points yields that the temperature and tearing energy satisfy a power-of-three relationship, as in Equation (29):(29)$$f\left(t\right)=a_{1}t^{3}+a_{2}t^{2}+a_{3}t+a_{4}$$
where t is the temperature, $a_{1}$, $a_{2},a_{3}$, and $a_{4}$ are the relevant constants. Thus the equation for correcting the tearing energy, the
(30)$$T_{max}=\frac{2\pi lE_{0}f(t)}{\sqrt{\lambda }}$$

The following are fitted in this paper at 30%, 50%, 80% and 100% strain with critical tearing energy respectively as shown below:(31)$$f\left(t\right)=-3.128e^{-6}t^{3}+3.810e^{-4}t^{2}-0.016t+0.985$$
(32)$$f\left(t\right)=-3.251e^{-6}t^{3}+3.911e^{-4}t^{2}-0.016t+0.982$$
(33)$$f\left(t\right)=-6.316e^{-6}t^{3}+7.214e^{-4}t^{2}-0.026t+0.975$$
(34)$$f\left(t\right)=-7.501e^{-6}t^{3}+8.576e^{-4}t^{2}-0.031t+0.970$$
(35)$$f\left(t\right)=-3.963e^{-6}t^{3}+4.430e^{-4}t^{2}-0.020t+0.971$$

Thus, the rubber thermodynamic coupling fatigue model can be obtained as
(36)$$N=\int_{l_{0}}^{l_{c}}B^{-1}T^{-F}da=\frac{1}{B\left(F-1\right){\left(\frac{2\pi f(t)}{\sqrt{\lambda }}E_{0}\right)}^{F}}(\frac{1}{l_{0}^{F-1}}-\frac{1}{l_{c}^{F-1}})$$

An interesting phenomenon was also found in the experimental observation of crack extension paths. In the range of 30–100% of strain level, the crack paths of crack extension experiments at room temperature were more regular, basically expanding perpendicular to the direction of loading, while the extension paths at high temperature were more tortuous, usually zigzagging forward. The possible reason is that the temperature of the rubber crack surface is higher in high temperature environment, and the surface molecules are more easily activated, which accelerates the rate of crack expansion.

### 3.7. Validation of Fatigue-Thermal Coupling Model

The hyperelastic intrinsic model and fatigue life prediction model of rubber were established in the previous section. In order to ensure the accuracy of the subsequent tire simulation calculation, the prediction effect of the constructed model needs to be verified, so the fatigue life of the specimen at 50 °C was predicted by establishing the finite element model of the planar tensile specimen and compared with the Thomas model and experimental results to verify the prediction effect of the constructed thermodynamic coupling model, as shown in Figure 22 is the finite element model of the plane tensile specimen. Figure 23 shows the loading amplitude curve.

The stress–strain clouds at 50% strain under planar tension are shown in Figure 24.

The calculated ODB file is imported into Endurica2020 software to calculate its fatigue life, and the Thomas model in the software library is selected for the calculation of life, and the finite element calculation result of life is shown in Figure 25, which is 8.315 × 10^5^ times.

As shown in Figure 26, the cloud plot of rubber specimen life calculated by using the thermally coupled fatigue model built in the previous section, the result is 6.588 × 10^5^ times, and the experimental result is shown in Figure 27, which is measured as 6.42 × 10^5^ times, as shown in Table 6, the deviation between the predicted result of Thomas model and the actual one is 29.5%, and the error between the predicted result of the thermally coupled fatigue model built in this paper and the experimental result is 2.6%. It indicates that the calculation accuracy of the proposed thermodynamic coupling fatigue model is high and can meet the requirements of the subsequent tire fatigue life calculation.

## 4. Study of Fatigue Micromechanisms in Rubber

Many scholars, when studying fatigue, often characterize it by macroscopic features such as the length, type, and lifetime of cracks, while the damage of polymeric materials such as rubber often begins with the breakdown of the bonding between the atoms of polymer chains. And by Champy et al. [42], who studied the fatigue process of natural rubber, it was concluded that cracking starts with debonding of the agglomerates from the rubber matrix and cavities at the ends of the agglomerates, and Pérocheau et al. [43] also found the phenomenon of cavities at the ends of the agglomerates, and Le Cam, who studied [44] the fatigue life of carbon black-filled natural rubber, came to a similar conclusion that the agglomerates and rubber matrix debonded to produce cracks.

Therefore, it is known from the above studies that rubber products need a variety of fillers in the rubber matrix to meet different requirements for use, and the addition of fillers will inevitably form agglomerates, and when subjected to alternating loads, defects will be formed between the rubber matrix and the agglomerates, and these defects will gradually expand and gather to form microcracks, and the fatigue failure of rubber materials starts from these microcracks. From a large number of scholars’ studies, it is known that the size of the initial microcrack has a large influence on the fatigue life of rubber materials, and the size of the initial crack mainly depends on the size of the agglomerates and the aggregates of inorganic fillers such as ZnO, and observing the scale of the agglomerates by scanning electron microscopy (SEM) is the best way at present. This chapter attempts to verify whether the use of the crack extension method is consistent with reality by studying the surface morphology of rubber materials before and after fatigue, and to explore the connection between the macroscopic and microscopic aspects of fatigue.

### 4.1. Experiments and Discussion of Results

#### 4.1.1. Experiment

In order to visually observe the changes of surface morphology of rubber materials before and after fatigue, this experiment was first conducted using a fatigue test bench, and then the new surface generated by crack expansion of the experimental specimen was cut off; the specimen without fatigue test was also quickly cut with a blade to expose the surface, and then the prepared specimen was vacuum-treated and scanned by SEM instrument.

#### 4.1.2. Discussion of the Results

Figure 28 shows the fast cut surface of the specimen without fatigue experiments. From the photo with 10,000 times magnification, it can be seen that the fillers are all well dispersed, in which the rubber matrix is the continuous phase and the other fillers are the dispersed phase. And it can be seen from the figure that the infiltration of the filler and the rubber matrix is relatively good, the vast majority of the filler is wrapped in the rubber matrix, and only a small part of the filler particles are not wrapped by the rubber, which is the source of the fatigue cracks in the rubber material.

Figure 29 shows the surface of the specimen after fatigue photographed with three-dimensional morphology, and it can be found that there are both relatively smooth and rough areas on the surface, and there are obvious fatigue streaks, which are obvious characteristics of ductile materials. These two kinds of areas were also observed when Yanhong Wang studied the fatigue phenomenon. This is due to shear fatigue experiments, the tearing energy used is much less than the critical tearing energy, so it has not reached the degree of instantaneous damage, and the fracture surface caused by this slow tearing is generally rough; and when expanding along the rough surface, the crack is not expanding uniformly in one direction, so it will cause stress concentration, and when the stress exceeds the range that the rubber can withstand, it will tear instantly thus producing a smooth area. Also from the figure, we can see that the longer the fatigue life, the more stripes and the more obvious the grooves are, which means the crack expansion rate is low.

Figure 30 is a 100-fold SEM photograph of the fatigue surface and the cut-off surface of the specimen without fatigue experiments, we can see that the surface before and after fatigue is very different, the surface without fatigue experiments is very smooth and has a beach-like ripple, which is a sign of cut-off; while the fatigue interface can be The surface without fatigue test is very smooth and has beach-like ripples, which is a sign of cut-off; while the fatigue interface can be seen to have peeling phenomenon, which is layer by layer overlapping, which is the rough surface formed by long time shear fatigue.

Figure 31 shows the SEM scanning photos of the unfatigued specimens. It can be observed that the fatigue-free interface is smooth, but there are pits. This is because the rubber specimen hardly stretches when it is cut off, so when the material is cut off, the filler sticks to the fracture surface.

Figure 32 is a 5000-fold scanning photo of SEM. It can be seen that the combination of filler and rubber is good, and many irregular gullies and holes are produced after fatigue. This structure is helpful to improve the fatigue life of rubber, because it is a process of generating new surfaces and can consume some energy.

Figure 33a shows a scanning photo of the surface of the rubber specimen cut off with a blade after it has been stretched to 200% strain for six times, and (b) shows a scanning photo of the surface of the specimen that has not been stretched. It can be seen that (b) the surface is smooth and the bonding degree between the filler and the rubber matrix is good, but after the Mullins effect is eliminated, it can be found that some fillers have been separated from the rubber matrix, indicating that the network formed by the filler and the rubber has begun to be destroyed.

### 4.2. Micromechanism of Rubber Fatigue

From a classical mechanical calculation—the stress problem of a perforated flat plate—it is known that a round hole, i.e., a defect in a flat plate, generates a stress concentration in the place where the defect exists, and when the stress exceeds the critical value, this becomes the initial part of the damage. Similarly, the fatigue damage of rubber is also a concentration of stress at the defective part, which causes the molecular chain to break. The three-dimensional cross-linked network is produced when the rubber is vulcanized, and the molecular chain with low bond energy is destroyed first when the stress is concentrated. The stress condition at the crack tip of the rubber material is an important factor influencing its crack expansion. The shear fatigue test presets the crack to simulate the expansion process of the material after it is cracked. Rubber material is a weak but tough material, and it can be observed that it can have a large deformation with a small force when it is slowly stretched. When the crack tip becomes blunt, then its ability to resist tearing becomes stronger, because at this time the stress is dispersed in several directions, so the reason why rubber can still bear the load when there is damage is the blunting of the crack tip.

Fatigue damage of rubber is actually the result of both physical and chemical damage. When the rubber material is loaded, the chain segments between the cross-linked bonds are oriented, all chain segments are stressed, and the chain segments are Gaussian-distributed, so the stress is also Gaussian-distributed. When the rubber is under a low load and has a small deformation, the conformation of the cross-linked network is sufficient to withstand the current deformation, so it is almost not destroyed, but as the deformation increases, the molecular chain is continuously stretched, and when the maximum deformation is reached, it cannot meet the current deformation. This causes the molecular chain to break and release the strain energy, which is partly used to overcome the new surface energy and partly transferred to the adjacent cross-linked network. Therefore, these adjacent cross-linked networks will also be damaged by the excessive energy. Many fillers are added to rubber products, and when the particle size of these inorganic particles is larger than microns, their interface with the rubber matrix is a defect. When rubber materials are subjected to alternating loads, damage is actually produced in the first cycle, as demonstrated in the previous sections on the Mullins effect. After experiencing the first tension cycle, many microdefects have formed inside the rubber, but the effect of the first few tensile loads on the fatigue life is minimal due to the fact that when the loading is stopped without causing macroscopic visible cracks. These microcracks cause physical–chemical effects, such as infiltration and diffusion, but will be repaired automatically. After undergoing many cycles of loading, more and more molecular chains are damaged, which eventually leads to visible cracks. So one way to improve the fatigue life of rubber is to improve the dispersion of the filler in the rubber.

Fatigue damage often does not occur immediately, but rather over a long period of time since it involves chemical reactions. Chemical reactions generally refer to the erosion of rubber molecular chains by gases, the most important of which is to react with oxygen. When using the universal tensile machine in accordance with national standards for rubber specimens in the tensile test, it was found that the results in the air environment and nitrogen environment are essentially the same, which also shows that chemical reactions need a certain amount of time to occur. A single stretch in oxygen cannot immediately cause it to react with the rubber as the process is not instantaneous. But in different gas environments for fatigue experiments, the results are very different. Sainter studied the fatigue life of styrene–butadiene rubber in air and oxygen environments and found that the fatigue life in air was eight times higher than that in oxygen. In the presence of oxygen, cross-linking and chain breaking occur simultaneously, and the more oxygen content, the more violent the reaction. When cross-linking is predominant, the rubber will harden and become brittle, eventually leading to fracture, and if chain breaking is predominant, then the rubber will become soft and eventually be destroyed because the strength becomes smaller.

## 5. Conclusions

In this paper, the rubber fatigue test platform was built. The thermal coupling fatigue prediction model of rubber was established based on a large number of experimental data, and the microscopic mechanism of rubber fatigue was analyzed. The main contents include:

(1) By conducting Mullins effect experiments on different formulations of vulcanized rubber, it is found that the Mullins effect can be eliminated after 5–6 stretches.

(2) The effect of different stretching speeds on the stress-strain curves was investigated and it was found that as the stretching speed increased, the stress required to achieve the same strain also increased.

(3) By conducting fatigue experiments under different working conditions, it was found that the life is longer when there is a stress ratio, and the life continues to increase as the stress ratio increases; the effect of the loading method on the crack extension rate is negligible; frequency itself has no effect on fatigue, high frequency makes the rubber rubber temperature rise, thus accelerating the crack extension; the effect of the Mullins effect on the fatigue life of rubber needs to be verified by subsequent system experiments to verify;

(4) By fitting the experimental data, it was found that temperature and tearing energy could fit a power-of-three function, so the influence of temperature was considered in the calculation of tearing energy, and a rubber thermodynamic coupling fatigue prediction model was established; the fatigue life of the specimen at high temperature was calculated using the thermodynamic coupling model, and the error was 2.6% compared with the experimental results, which verified the accuracy of the model;

(5) The surface morphology of rubber before and after fatigue was observed by SEM, and it was concluded that the source of crack extension was the debonding of rubber matrix and agglomerates; from the perspective of molecular mechanism, it was explained that the process of rubber fatigue damage was actually the process of overloading the cross-linked network and thus being destroyed.

## Figures and Tables

**Figure 1 polymers-15-02746-f001:**
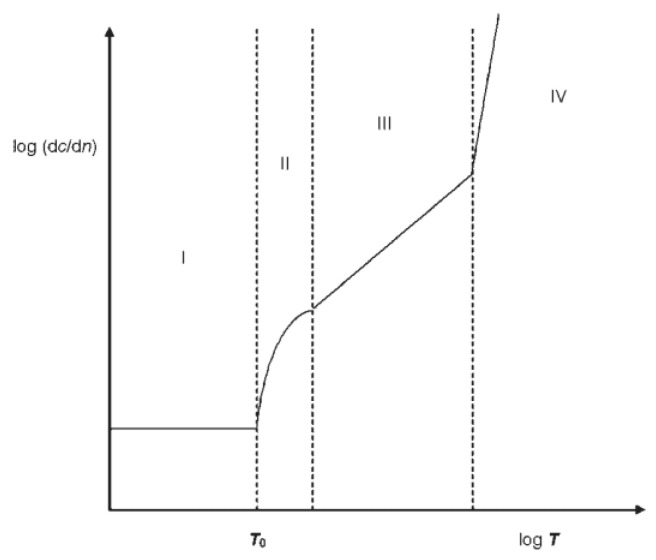
Crack expansion model of rubber material.

**Figure 2 polymers-15-02746-f002:**
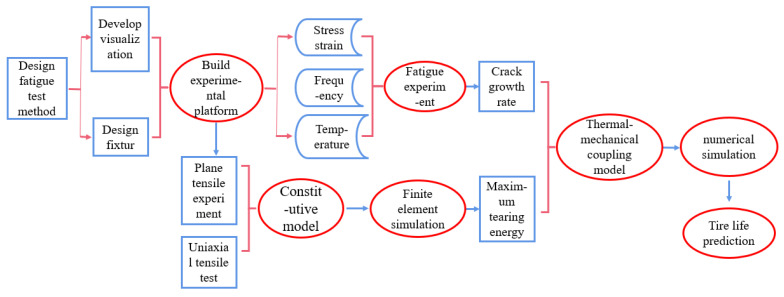
Research Route.

**Figure 3 polymers-15-02746-f003:**
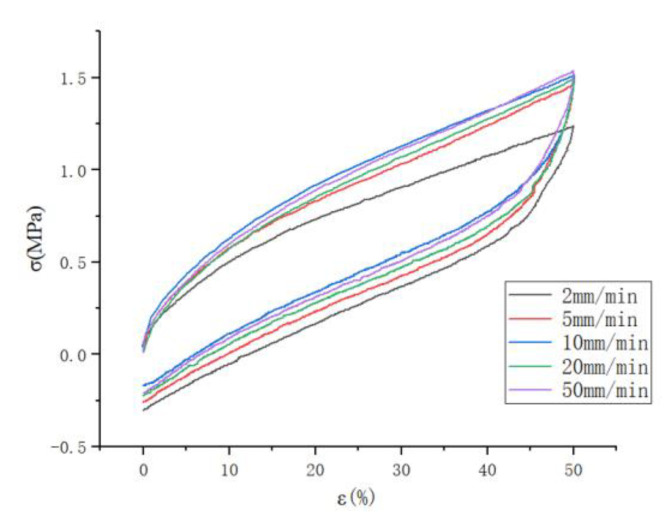
Tensile curves at different tensile speeds with strain levels of 50%.

**Figure 4 polymers-15-02746-f004:**
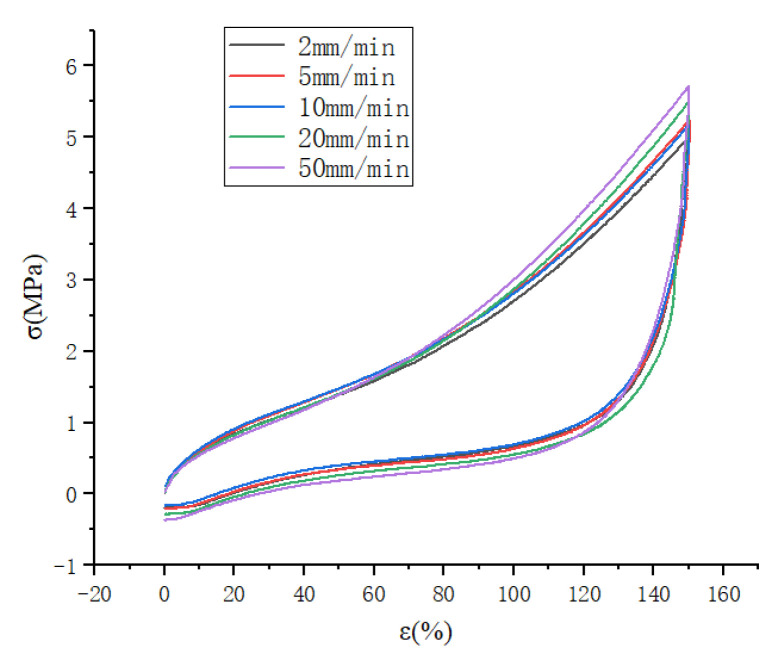
Stretching curves at different stretching speeds with 150% strain level.

**Figure 5 polymers-15-02746-f005:**
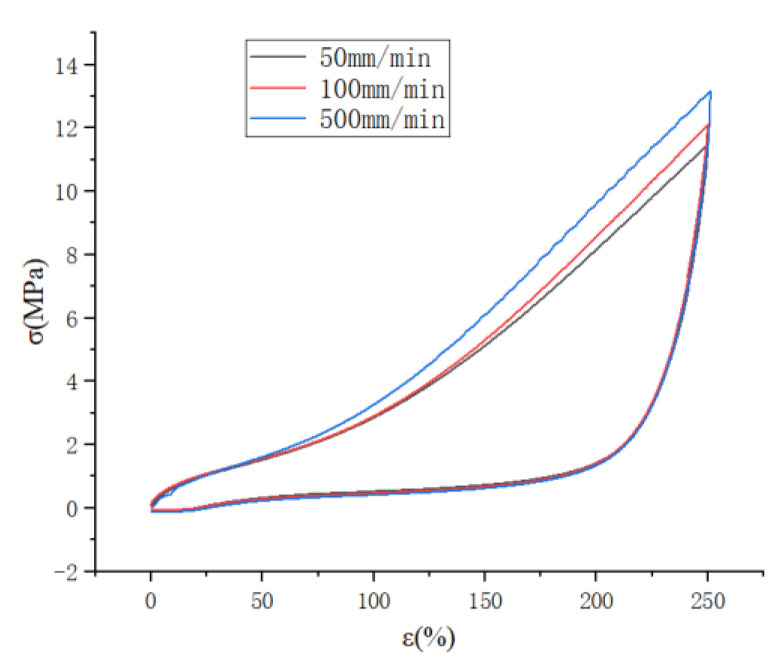
Cyclic curves for different stretching speeds at 250% strain level.

**Figure 6 polymers-15-02746-f006:**
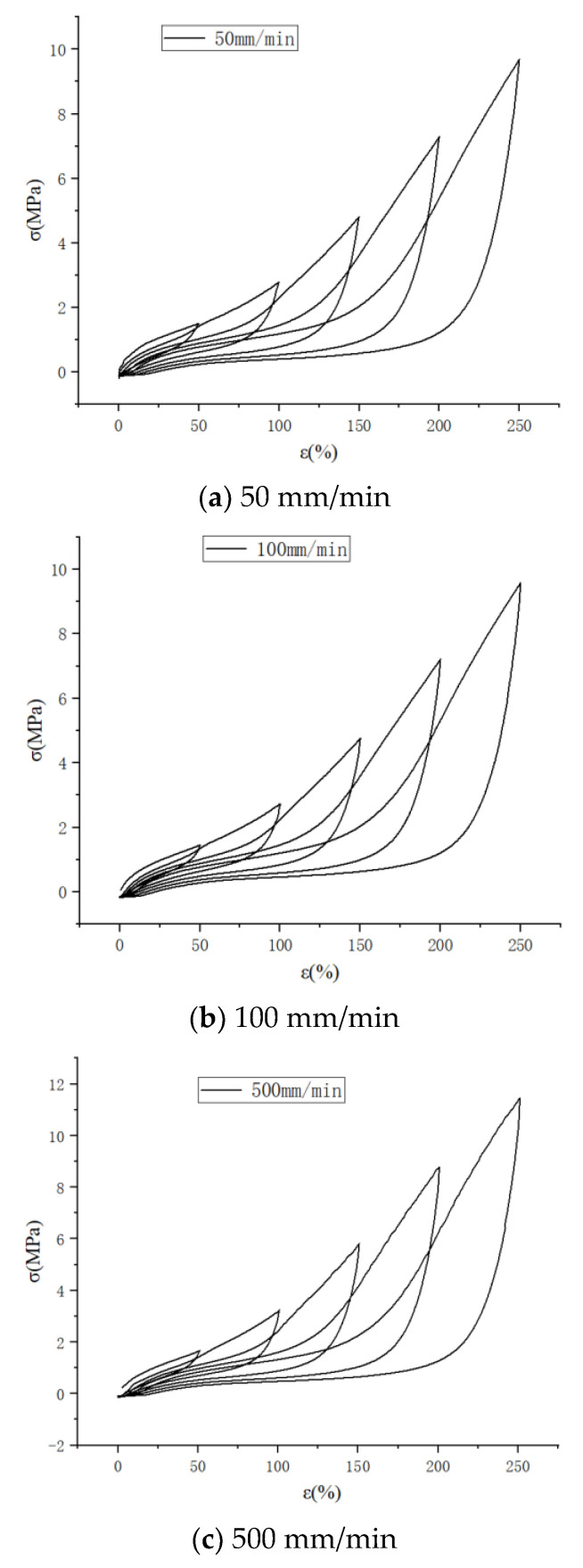
Uniaxial cyclic stretching curves at different stretching speeds.

**Figure 7 polymers-15-02746-f007:**
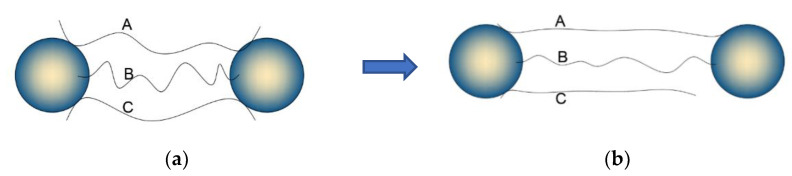
Rubber molecular chain breakage model. (**a**) the chain is not stretched. (**b**) the chain is stretched. A, B, and C are molecular chains.

**Figure 8 polymers-15-02746-f008:**
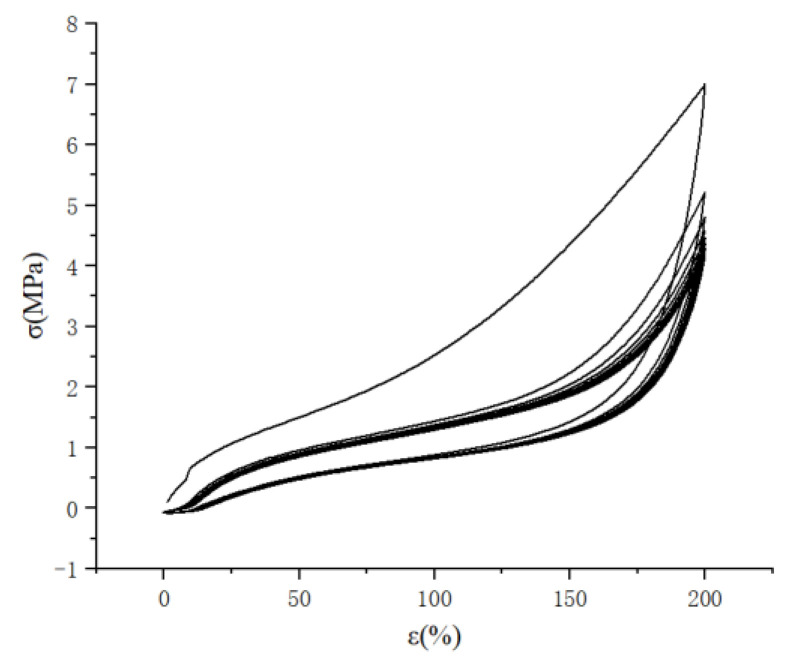
Graph of cyclic tensile stress-strain curve of rubber material.

**Figure 9 polymers-15-02746-f009:**
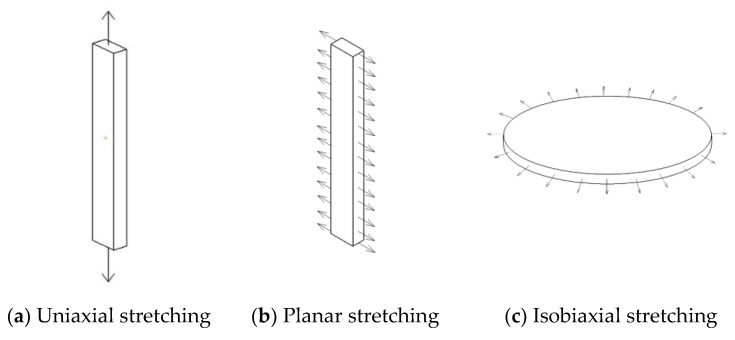
Loading diagram of uniaxial tension, plane tension and equal biaxial tension.

**Figure 10 polymers-15-02746-f010:**
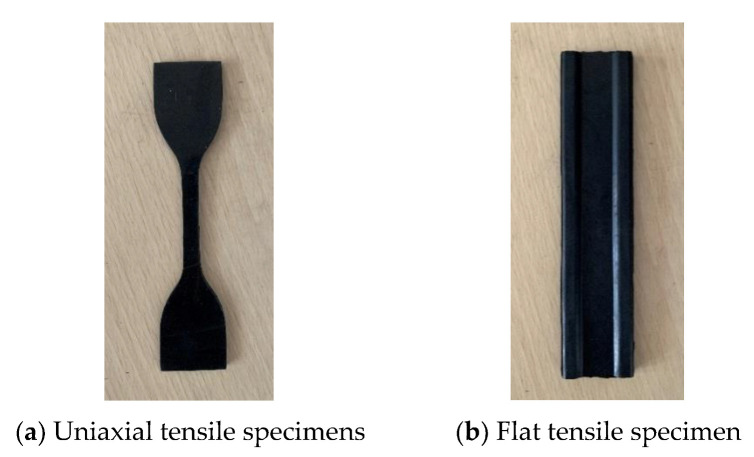
Uniaxial tensile specimen and plane tensile specimen.

**Figure 11 polymers-15-02746-f011:**
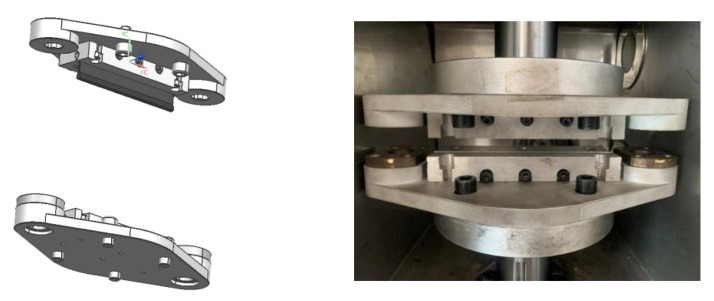
New fixture.

**Figure 12 polymers-15-02746-f012:**
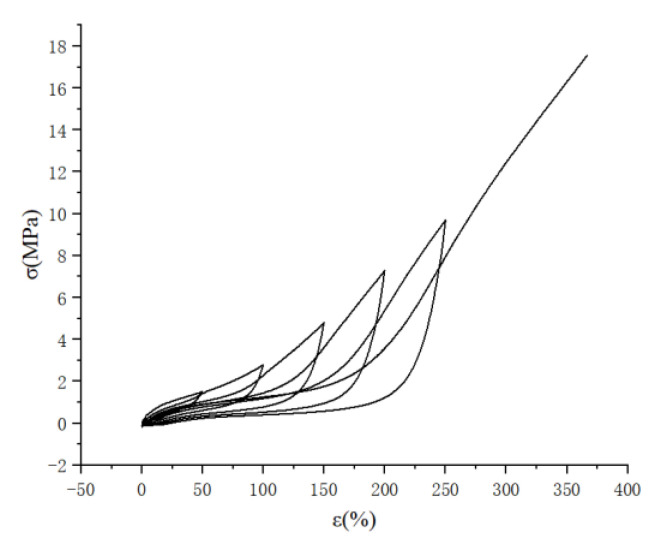
Stress-strain curve.

**Figure 13 polymers-15-02746-f013:**
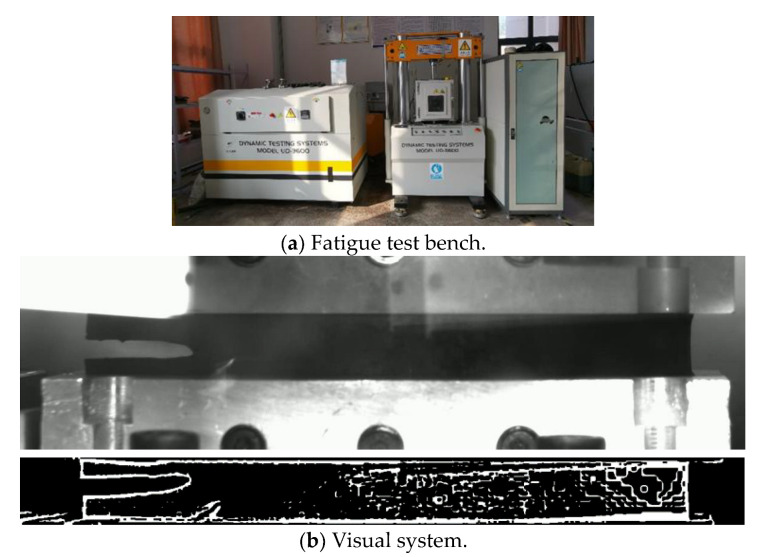
Dynamic fatigue analysis system.

**Figure 14 polymers-15-02746-f014:**
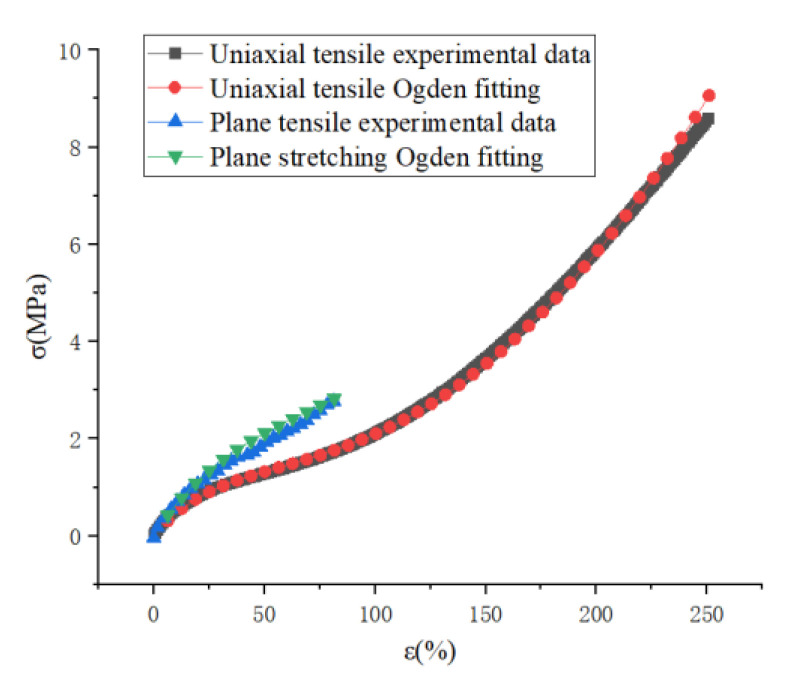
Uniaxial tensile and planar tensile experimental data and fitted data.

**Figure 15 polymers-15-02746-f015:**
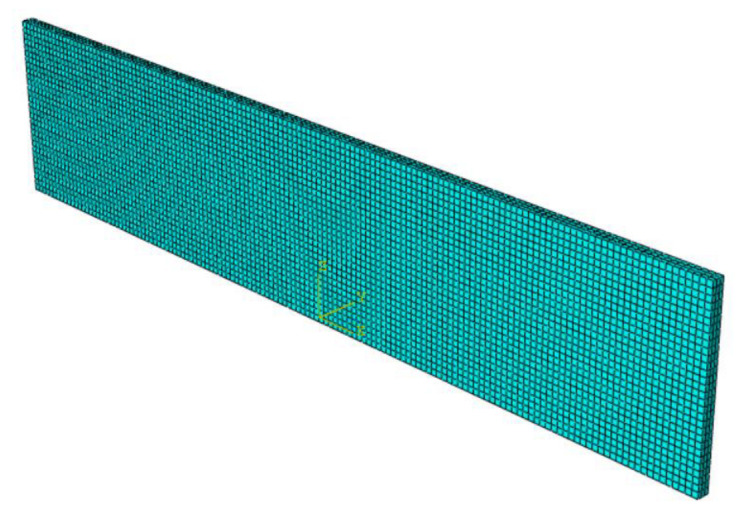
Finite Element Model Grid Diagram.

**Figure 16 polymers-15-02746-f016:**
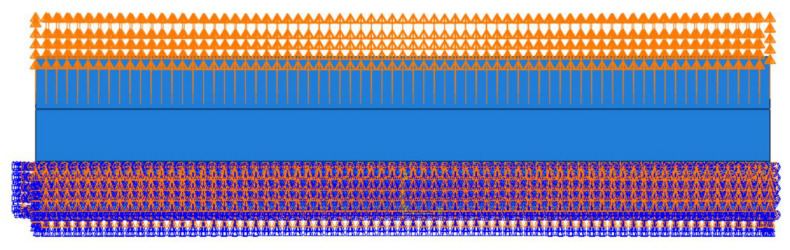
Loading Mode.

**Figure 17 polymers-15-02746-f017:**
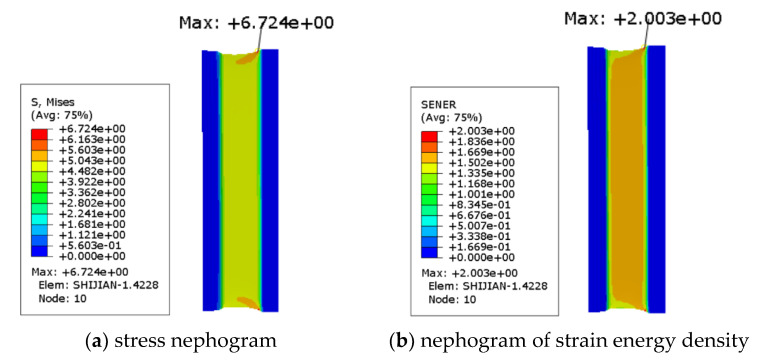
Simulation Results of Stress and Strain Energy Density.

**Figure 18 polymers-15-02746-f018:**
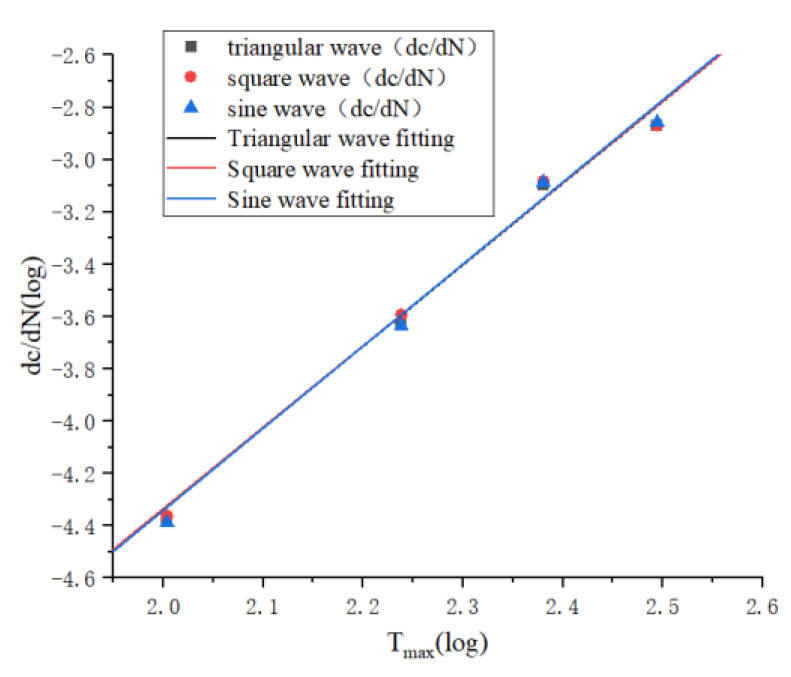
Fitting curves of tear energy and crack expansion rate under different loading methods.

**Figure 19 polymers-15-02746-f019:**
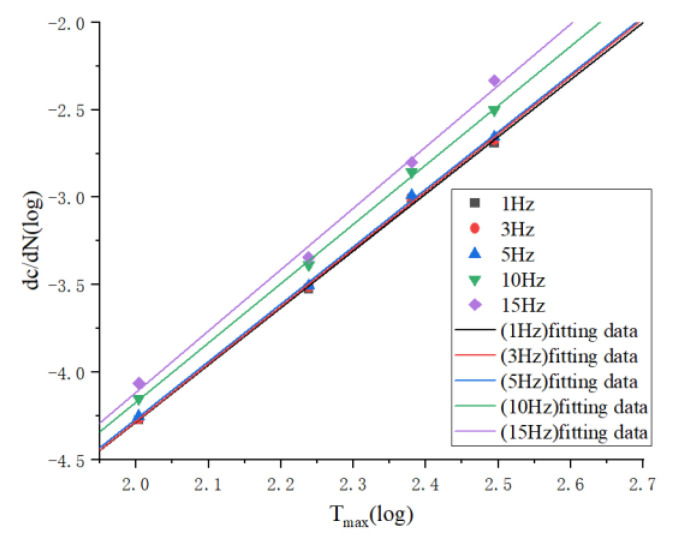
Fitting curves of crack expansion rate and tearing energy at different frequencies.

**Figure 20 polymers-15-02746-f020:**
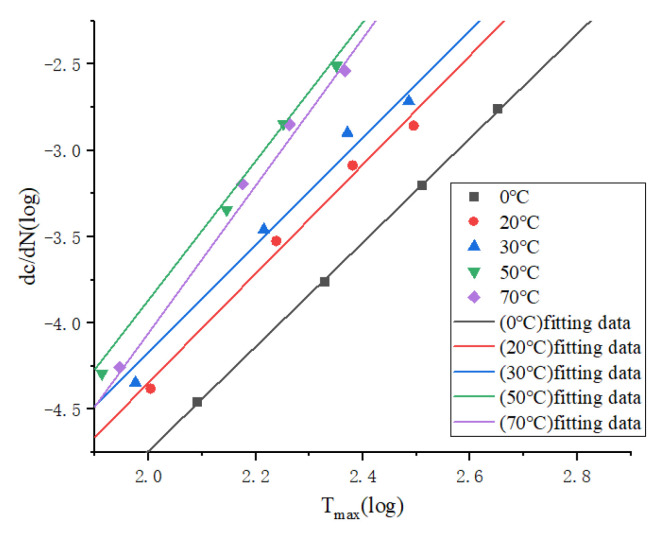
Fitting curves of crack expansion rate and tearing energy at different temperatures.

**Figure 21 polymers-15-02746-f021:**
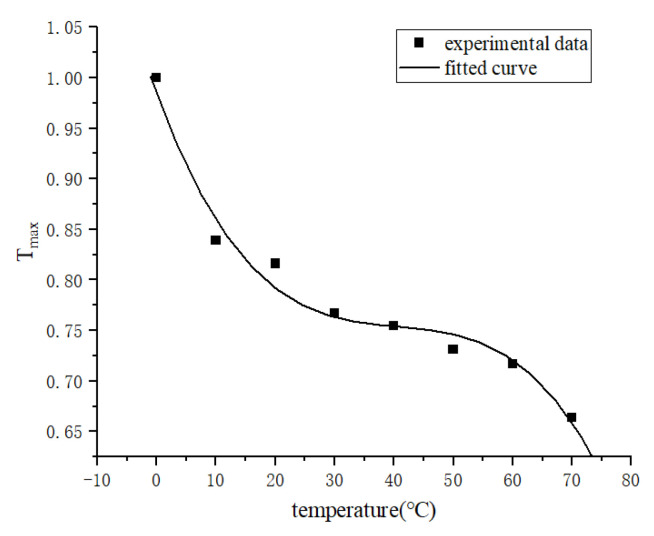
Tearing energy versus temperature.

**Figure 22 polymers-15-02746-f022:**
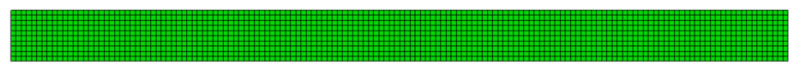
Finite element model of the planar tensile specimen.

**Figure 23 polymers-15-02746-f023:**
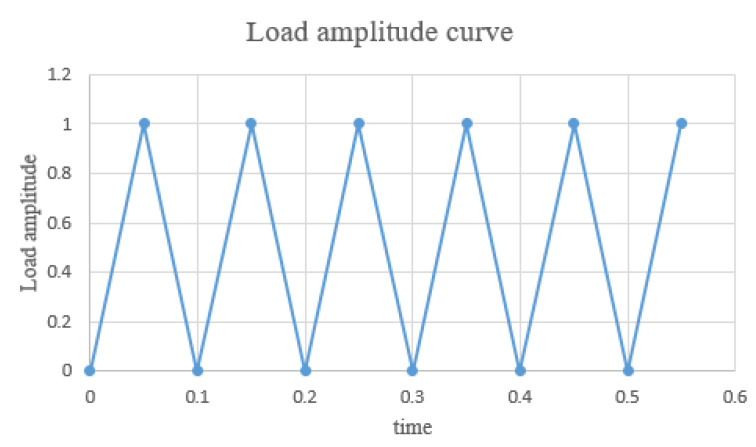
Loading amplitude curve.

**Figure 24 polymers-15-02746-f024:**
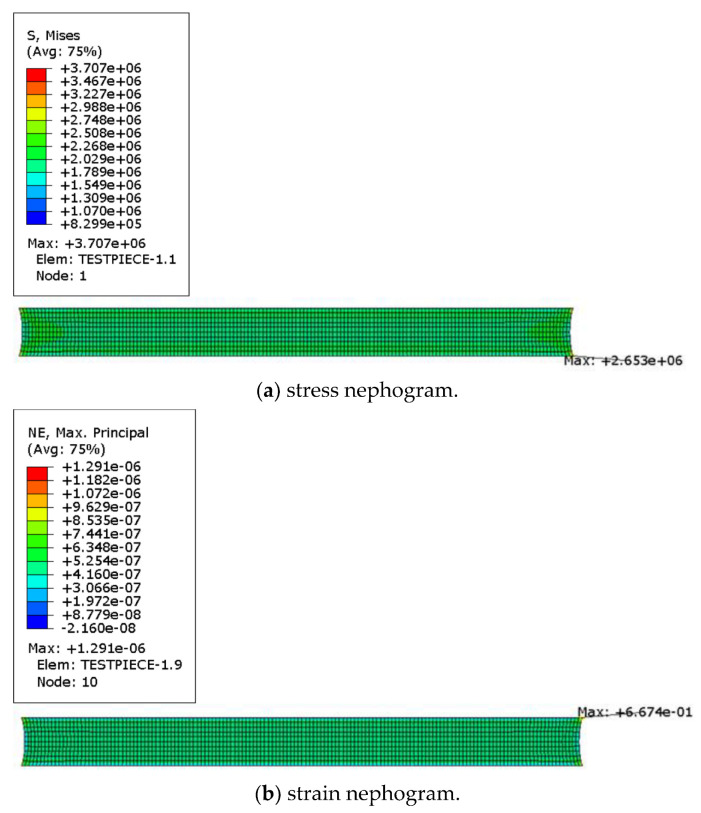
Stress-strain cloud of plane tensile specimen.

**Figure 25 polymers-15-02746-f025:**
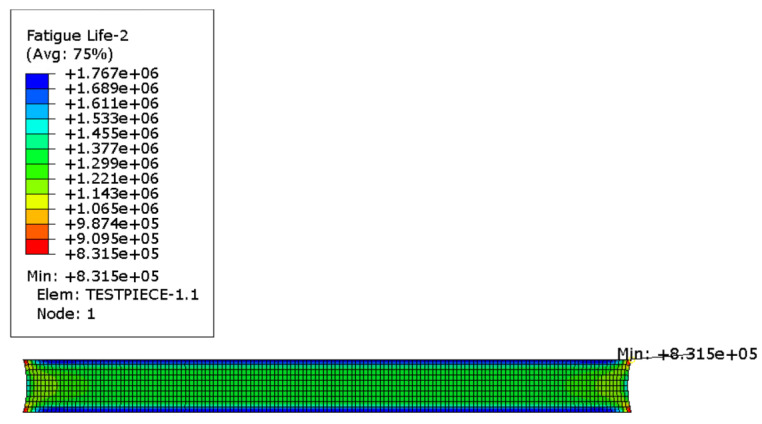
Life nephogram of plane tensile specimen.

**Figure 26 polymers-15-02746-f026:**
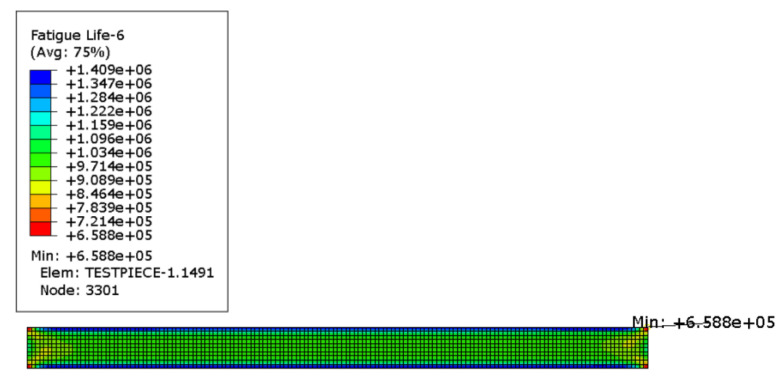
Nephogram of fatigue life of specimen under thermal-mechanical coupling.

**Figure 27 polymers-15-02746-f027:**
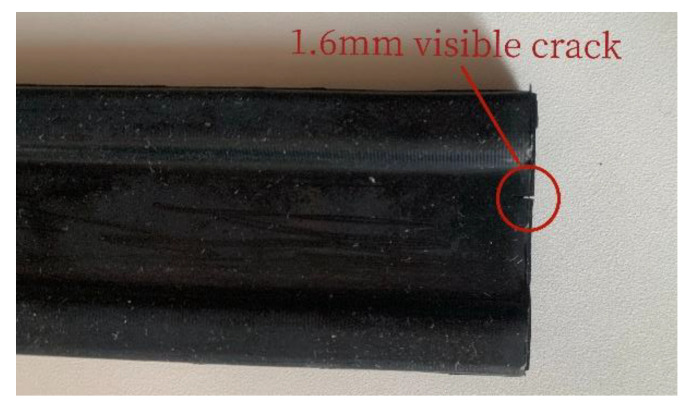
Fatigue specimen life.

**Figure 28 polymers-15-02746-f028:**
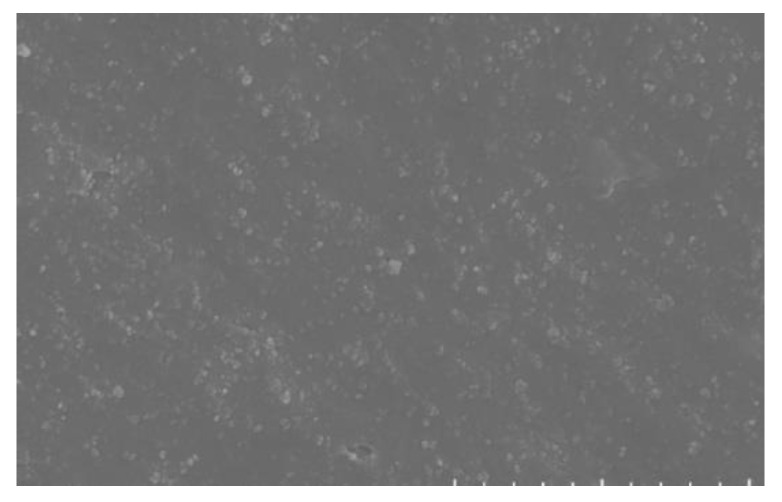
SEM10000 times morphology of rubber surface.

**Figure 29 polymers-15-02746-f029:**
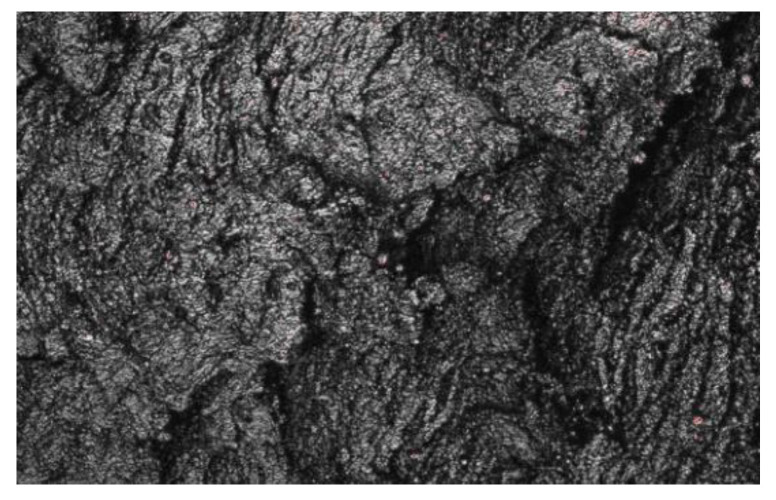
Three-dimensional photos of fatigue surface.

**Figure 30 polymers-15-02746-f030:**
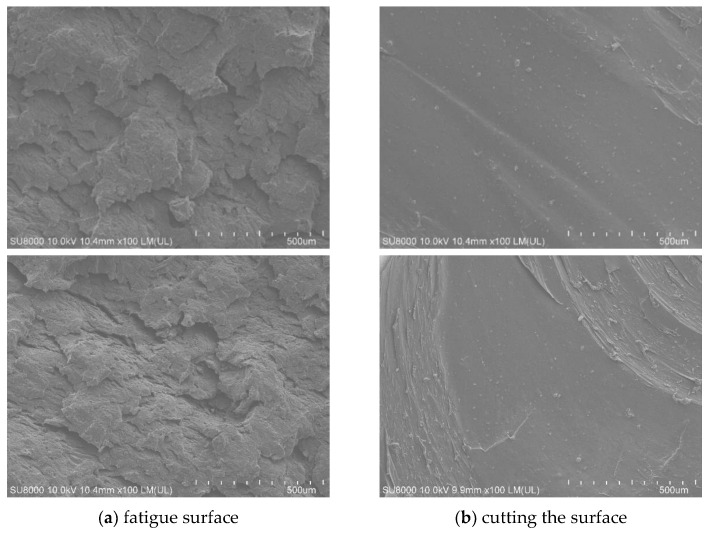
Morphology of Rubber Specimen under SEM100 times.

**Figure 31 polymers-15-02746-f031:**
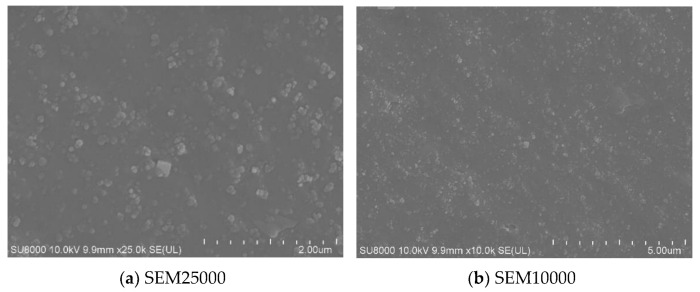
SEM surface morphology of cut-off surface.

**Figure 32 polymers-15-02746-f032:**
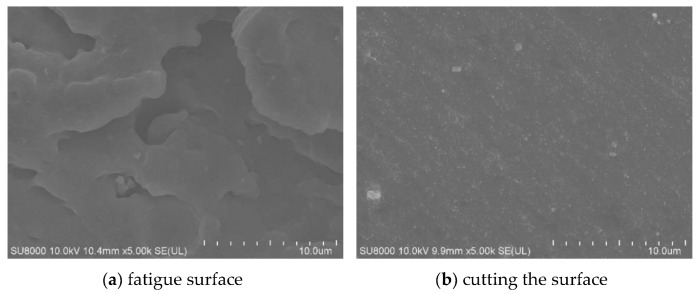
Morphology of Rubber Specimen at 5000 times of SEM.

**Figure 33 polymers-15-02746-f033:**
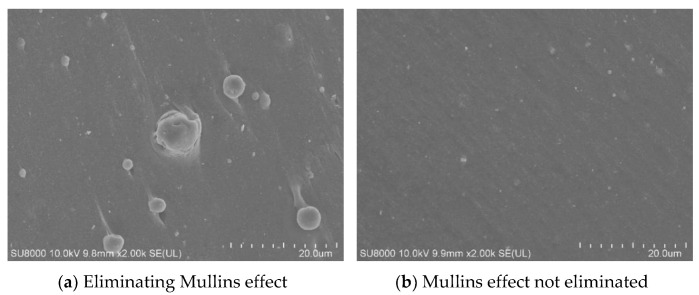
Shape of rubber specimen at SEM2000 times.

**Table 1 polymers-15-02746-t001:** Material parameters fitted by Ogden model.

No.	MU1	MU2	MU3	ALPHA1	ALPHA2	ALPHA3
Parameters	1.485	−5.016	10.038	1.704	12.501	−24.999

Note: The unit of each parameter is MPa.

**Table 2 polymers-15-02746-t002:** Strain energy density at different strain levels.

Strain	30%	50%	80%	100%
strain energy density	0.745	1.385	2.311	3.370
Tearing energy	102.637	177.633	270.572	374.313

Note: The unit of strain energy density is mJ/mm^3^.

**Table 3 polymers-15-02746-t003:** Crack expansion rates at different stress ratios.

No.	R	Frequency	Waveform	Strain	*dc*/*dN*
1	0	4 Hz	Sine wave	80%	6.143 × 10^−4^
2	1/3	4 Hz	Sine wave	80%	6.944 × 10^−5^
3	1/2	4 Hz	Sine wave	80%	3.472 × 10^−5^

Note: The unit of crack expansion rate is mm/c.

**Table 4 polymers-15-02746-t004:** Crack extension rate under different orientations.

No.	Orientation	Strain	Waveform	*dc*/*dN*
1	Vertical	50%	Sine wave	2.778 × 10^−4^
2	Parallel	50%	Sine wave	3.543 × 10^−4^

Note: The unit of crack expansion rate is mm/c.

**Table 5 polymers-15-02746-t005:** Effect of Mullins effect on crack expansion rate.

No.	Mullins Effect	Strain	Waveform	*dc*/*dN*
1	Eliminated	50%	Sine wave	2.454 × 10^−4^
2	Not eliminated	50%	Sine wave	2.315 × 10^−4^

Note: Eliminating the Mullins effect means stretching the specimen to 200% strain for 5 times and then presetting the crack for the fatigue test.

**Table 6 polymers-15-02746-t006:** Comparison of life prediction effects.

Different Models	Thomas Model	Thermodynamic Coupling Model	Experimental Results
Fatigue life	8.315 × 10^5^	6.588 × 10^5^	6.42 × 10^5^

## Data Availability

Not applicable.

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
