# Peer review of "Development of Fatigue Life Model for Rubber Materials Based on Fracture Mechanics"

_polymers, 2023, doi:10.3390/polym15122746_

Round 1

Reviewer 1 Report

After a comprehensive review of the manuscript " Development of fatigue life model for rubber materials based 2 on fracture mechanics", Topic and paper are interesting. There is no doubt in the originality of the work but I think the manuscript should  be improved.  I'll try to clear my point through the following comments:

1.      In the abstract, you must give results of failure modes, load-carrying capacities, energy dissipation capacities, peak amplitudes, fatique strength, etc. % are needed. The introduction should be improved. More general literature review is needed and novelty of the research should be stated clearly. I think it should be add a new section which is “research significance”. In other words, the novelty and importance of the research should be clarified using up-to-date researches around the globe.

2.      The differences from previous studies must be specified clearly. Experiences from all over the world could help authors for performing a comprehensive research. You may also add flow chart about your study.

3.      What is the limit for numerical analysis?

4.      What is the criterion of authors for choosing the tested specimens? Using different specimens or even different scales could jeopardize the results of the research or not? I think it is better to choose these specimens from a constructed or at least a designed in  real life.

5.      There are figure sequencing errors. should be revised. SEM analyzes should be detailed. It should be marked on the relevant figures.

6.      The conclusion part is only a summary of the results. We need technical and general advices here which can be used by others (both researches and engineers).

7.      The work is a good report of experiments investigation, but which are the lessons learnt? The authors have to clarify before acceptance.

Best regards.

Minor editing of English language required

Author Response

Dear Reviewer,

    We are very grateful for your comments regarding our manuscript . All your suggestions are very important to us, both for composing the manuscript and our further research. We have studied comments carefully and have made corrections which we hope meet with approval.

    For your third question, I think the advantages of numerical analysis are high efficiency and saving money, but the calculation accuracy depends largely on the accuracy of experimental parameters and calculation models.

    For the fourth question, When the aspect ratio of the specimen is greater than or equal to 10, it can be approximately considered that the stress state is in a pure shear state, so the size of the specimen is designed to be 140mm long, the height of the working area is 10mm, and the thickness is 2 mm. It is necessary to preset a crack of 25mm for fatigue test, because it is considered in mechanics that when the crack length is more than or equal to twice the height, the influence of edge effect on shear fatigue test can be ignored. Therefore, in theory, the specimens that meet the above conditions can be tested for shear fatigue.

Other suggestions have been revised in the manuscript, please refer to them. Thank you!

Kind regards!

Reviewer 2 Report

This paper talks about the development of fatigue life model for rubber materials based on fracture mechanics. The following points need to be clarified for paper to be considered for the publication:

  1. The paper is too long and difficult to be followed. Especially, the abstract and conclusion of the paper needs to be shortened significantly. Only the main points need to be mentioned.
  2. The paper can be divided into two parts, may be the temperature effect could be shown in a separate paper.
  3. The structure of the paper is different. Normally the experimental results are given first followed by simulation to describe the experimental results. In this paper, an opposite is done. Sections 3 and 4 were not linked to each other. The authors can not predict some features of the experimentally obtained results with the FE  model.
  4. It is suggested that the authors should modify the last part of Section 1 to: 1) clearly mention the goal and novelty of this work (explicitly mention what is not achieved before but done with your study), 2) mention the methodology used (Numerical, Experimental, etc)
  5. The description of the FE model with a figure showing the BCs, mesh, load, etc should be given.
  6. It would be better to validate the numerical model with the experiments with more than 1 data. However, only 1 data was considered in the study, see Table 3. Comparison of life prediction effects. To show the robustness of the developed models, the authors should consider this point. Fitting just single data does not mean the developed model is correct.
  7. Figure numbers inside the text do not match with the figure captions.
  8. The following articles related to fatigue could be added in the literature review for latest developments:

https://doi.org/10.1016/j.ijfatigue.2022.107034

https://doi.org/10.3390/polym15081949

9.      The paper lacks a discussion on the weakness and limitations of the present methodology/study.

  1. The conclusion part needs to be shortened just to highlight the main findings.

The language of the paper is ok.

Author Response

Dear Reviewer,

We are very grateful for your comments regarding our manuscript . All your suggestions are very important to us, both for composing the manuscript and our further research. We have studied comments carefully and have made corrections which we hope meet with approval.

    We have abridged the abstract and conclusion.
    For the third suggestion, our simulation uses the established thermo-mechanical coupling model to calculate the life, and compares the results with those of Thomas model to judge the accuracy of the thermo-mechanical coupling model. In the fourth section, the surface morphology was observed by SEM to explore the mechanism of rubber fatigue.

For other suggestions, we have revised them in the manuscript, please refer to them. Thank you!

Kind regards!

Round 2

Reviewer 1 Report

Thanks for the revision. 

Minor editing of English language required

Author Response

Dear reviewer,

I have revised the paper, please check it out!
Thank you!

Kind regards!

Reviewer 2 Report

The paper is revised. However, still, the conclusion is too long. Only the findings should be placed.

The language could be proofread.

Author Response

Dear reviewer,

I have revised the conclusion according to your suggestion, please refer to it.

Thank you!

Kind regards!
